# Symbolic Causal Inference via Operations on Probabilistic Circuits

**Benjie Wang**
Department of Computer Science
University of Oxford

**Marta Kwiatkowska**
Department of Computer Science
University of Oxford

## Abstract

Causal inference provides a means of translating a target causal query into a causal formula, which is a function of the observational distribution, given some assumptions on the domain. With the advent of modern neural probabilistic models, this opens up the possibility to perform accurate and tractable causal inference on realistic, high-dimensional data distributions, a crucial component of reasoning systems. However, for most model classes, the computation of the causal formula from the observational model is intractable. In this work, we hypothesize that probabilistic circuits, a general and expressive class of tractable probabilistic models, may be more amenable to the computation of causal formulae. Unfortunately, we prove that evaluating even simple causal formulae is still intractable for most types of probabilistic circuits. Motivated by this, we devise a conceptual framework for analyzing the tractability of causal formulae by decomposing them into compositions of primitive operations, in order to identify tractable subclasses of circuits. This allows us to derive, for a specific subclass of circuits, the first tractable algorithms for computing the backdoor and frontdoor adjustment formulae.

## 1 Introduction

The problem of causal inference is to estimate a given causal query on a data-generating system, given some assumptions on the system and available data generated from that system. The graphical framework of Pearl [15] provides an intuitive means of specifying these assumptions as a *causal graph* over observed variables, and a calculus [14] for translating a causal query into an expression involving probability distributions over observed variables (called a *causal estimand* or *formula*), given the causal graph. Consider, for example, the causal graphs in Figure 1. These represent causal assumptions on the domain; in particular, they show how observed variables are related qualitatively, and the presence of unobserved confounders (represented by bidirected arrows). Given these assumptions, the goal is then to estimate the *interventional distribution* $p_X(Y)$. For the graph in Figure 1a, this is given exactly by the well-known *backdoor adjustment* formula.

However, in practice, for complex, high-dimensional distributions $p$, there remain significant challenges to estimating (statistics of) the interventional distribution. The first challenge is to obtain accurate probabilistic models for the observational distribution, perhaps learned from data; fortunately, modern machine learning provides a plethora of expressive generative model classes. However, even given an *exact* model for the observational distribution, a second *computational* challenge remains: to compute the function given by the causal formula. For example, in the backdoor case, this involves computing an intractable summation/integral (exponential in the dimension of $Z$) of a product; as a result, we must resort to an approximate or heuristic algorithm such as a Monte Carlo estimate. For other causal formulae, the situation is even more complex as the approximation may have to be hand-designed and/or come with little or no guarantees; for example, for the frontdoor formula in Figure 1b, effective estimators have only very recently been developed [8, 11].

36th Conference on Neural Information Processing Systems (NeurIPS 2022).

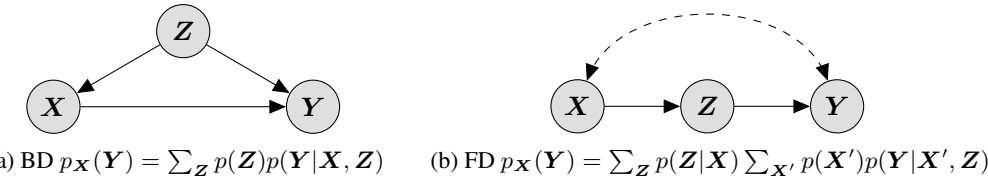

(a) BD $p_{\boldsymbol{X}}(\boldsymbol{Y}) = \sum_{\boldsymbol{Z}} p(\boldsymbol{Z}) p(\boldsymbol{Y}|\boldsymbol{X}, \boldsymbol{Z})$     (b) FD $p_{\boldsymbol{X}}(\boldsymbol{Y}) = \sum_{\boldsymbol{Z}} p(\boldsymbol{Z}|\boldsymbol{X}) \sum_{\boldsymbol{X}'} p(\boldsymbol{X}') p(\boldsymbol{Y}|\boldsymbol{X}', \boldsymbol{Z})$

Figure 1: Example causal graphs and their associated identifying formulae

In this paper, we propose an alternative question and approach to the computational problem: namely, are there interesting model classes for which *exact* computation of causal formulae is tractable? This question is of clear practical interest, as any such sufficiently expressive model class could be used as a representation to perform causal inference *reliably* with guarantees on the exactness of the result. As a logical starting point, we consider the framework of probabilistic circuits (PC) [3], a general class of probabilistic models which are notable for the ability to perform exact and efficient *probabilistic inference* with respect to the model. Unfortunately, our first result is that, for almost all existing types of probabilistic circuits, *causal inference* is #P-hard, even for the simplest case of backdoor adjustment. Nonetheless, we propose a method for analysing their tractability based on decomposing the causal formula via the *do-calculus*. This allows us to characterize a subclass of probabilistic circuits which admit tractable algorithms for computation of frontdoor and backdoor adjustment formulae; namely, *structured-decomposable* and *strongly deterministic* PCs.

**Related Work**   The relationship between probabilistic circuits and causality has its roots in the *compilation* methods in the seminal work of [5], which described an inference approach for (causal) Bayesian networks that involved compiling their graphs into tractable arithmetic circuits; subsequent work has further examined the relationship between compiled arithmetic circuits and causal models [1, 6, 2]. Recent trends, however, have increasingly moved towards *general* probabilistic circuits, which are defined by their *structural properties* and can be learned directly from data [17, 4, 21, 7]. However, obtaining an exact causal interpretation of general probabilistic circuits has proved much less successful [23, 13]. The only practical prior causal inference method for such circuits is the neural parameterization of Zecevic et al. [22], but this lacks exactness guarantees and is only applicable to fully observed settings. In contrast, our paper aims to characterize the complexity of and propose methods for *exact* causal inference for *general* probabilistic circuits.

## 2   Preliminaries

A probabilistic circuit $C$ over a set of variables $\boldsymbol{V}$ is a computational graph (rooted DAG) consisting of three types of nodes: leaf, sum and product nodes. The circuit encodes a function $C : \boldsymbol{V} \to \mathbb{R}_{\geq 0}$, which can be interpreted as a probability distribution. In particular, each leaf node $L$ represents a function over some subset $\phi(L)$ of the variables, each product node $P$ multiplies the functions given by its children, i.e., $P = \prod_{N_i \in ch(P)} N_i$, while each sum node $T$ is defined by a weighted sum of its children, i.e., $T = \sum_{N_i \in ch(T)} \theta_i N_i$. We assume (w.l.o.g.) in this paper that each product node has exactly two children. The weights $\theta_i \in \mathbb{R}_{\geq 0}$ of the sum nodes are referred to as the *parameters* of the PC. The *scope* of a node $N$ denotes the set of variables $N$ specifies a function over, and can be defined recursively for each product or sum node $N$ as $\phi(N) = \cup_{N_i \in ch(N)} \phi(N_i)$. Finally, the *size* of a circuit, denoted $|C|$, is defined to be the number of edges in the circuit.

The tractability properties of probabilistic circuits depend on the structural properties they satisfy. A PC is *decomposable* if the children of a product node have distinct scopes (and thus partition the scope of the product node), is *smooth* if the children of a sum node have the same scope, and is *deterministic* if, for every instantiation $\boldsymbol{w}$ of the scope of a sum node, at most one of its children $N_i$ evaluates to a non-zero value $N_i(\boldsymbol{w})$. Decomposability and smoothness together enable tractable marginal inference; that is, for any subset $\boldsymbol{W} \subseteq \phi(N)$ of the scope of a node $N$, we can compute $N(\boldsymbol{W}) := \sum_{\phi(N) \backslash \boldsymbol{W}} N(\phi(N))$ efficiently. In this paper, we will need an additional property, called *structured decomposability* [16, 12], which intuitively means that the scopes of all product nodes in the PC decompose (partition) in the same way. More formally, a PC is structured decomposable if it respects some vtree $v$. A vtree $v = (M, E)$ for a set of variables $\boldsymbol{V}$ is a full, rooted binary tree

with nodes $M$ and edges $E$ whose leaves are in one-to-one correspondence with the variables in $\boldsymbol{V}$. For any leaf node $m$ in the vtree, we define the scope $\phi(m)$ to be the singleton set containing the corresponding variable, and for any other node, we define $\phi(m) = \phi(m_L) \cup \phi(m_R)$, where $m_L, m_R$ are the left and right children of $m$. A PC $C$ respects a vtree if every product node $P$ in $C$ has a node $m$ in the vtree with the same scope, and decomposes (partitions) in the same way. We show an example of a structured decomposable, smooth, and deterministic circuit in Figure 3a.

## 3 Tractability of Backdoor Adjustment

In this section, we begin by considering the basic case of computing causal effects of the form $p_{\boldsymbol{x}}(\boldsymbol{y})$, where $\boldsymbol{X}, \boldsymbol{Y} \subseteq \boldsymbol{V}$ are disjoint subsets of the observed variables, and $\boldsymbol{x}, \boldsymbol{y}$ are instantiations of $\boldsymbol{X}, \boldsymbol{Y}$. One of the most common cases where the causal effect is identifiable is when there exists a valid *backdoor adjustment set* $\boldsymbol{Z} \subseteq \boldsymbol{V} \setminus (\boldsymbol{X} \cup \boldsymbol{Y})$ (also known as the conditional exchangability/ignorability assumption). In particular, $\boldsymbol{Z}$ is a valid adjustment set iff it satisfies a graphical condition known as the *backdoor criterion* with respect to $\boldsymbol{X}, \boldsymbol{Y}$ on the causal diagram $G$. Whenever such a set exists, the causal effect $p_{\boldsymbol{x}}(\boldsymbol{y})$ is given by the backdoor adjustment formula

$$p_{\boldsymbol{x}}(\boldsymbol{y}) = \sum_{\boldsymbol{z}} p(\boldsymbol{z}) p(\boldsymbol{y}|\boldsymbol{x}, \boldsymbol{z}) \tag{1}$$

Now suppose that we have a probabilistic model $M$ representing the distribution $p$ over observed variables, i.e. $M(\boldsymbol{V}) \equiv p(\boldsymbol{V})$, perhaps learned from data. Despite the apparently simplistic setup, computing (1) is not straightforward for many probabilistic models $M$. In particular, notice that both the expressions $p(\boldsymbol{z})$ and $p(\boldsymbol{y}|\boldsymbol{x}, \boldsymbol{z})$ require the other variables in the model to be marginalized out, which is generally not tractable for probabilistic models. Notably, however, even if the model allows for tractable marginal evaluation, and thus the computation of the probabilistic expressions for specific values $(\boldsymbol{x}, \boldsymbol{y}, \boldsymbol{z})$ of $(\boldsymbol{X}, \boldsymbol{Y}, \boldsymbol{Z})$, the summation is computationally intractable as it takes exponential time in the dimension of $\boldsymbol{Z}$.

It is thus a natural question whether there exist interesting classes of probabilistic models for which causal inference is tractable. A natural starting point is to look at (decomposable and smooth) tractable probabilistic circuits, but we have seen that tractable marginal inference alone is not sufficient. The tractability of answering causal inference queries *exactly* for probabilistic circuits, where exactness is a hallmark and extremely desirable property of probabilistic circuit queries, has remained an important open question. Unfortunately, we answer in the negative in the following result:

**Theorem 1.** *The backdoor query for decomposable and smooth circuits is #P-hard, even if the circuit is structured decomposable and deterministic.*

This implies hardness of causal inference for PCs whenever there is a valid backdoor adjustment:

**Corollary 1.** *For any causal effect $p_{\boldsymbol{x}}(\boldsymbol{y})$ and causal diagram $G$ such that the query is identifiable through a backdoor adjustment, and the observational distribution $p(\boldsymbol{V})$ given as a decomposable and smooth circuit $C(\boldsymbol{V}) \equiv p(\boldsymbol{V})$, computing $p_{\boldsymbol{x}}(\boldsymbol{y})$ is #P-hard, even if the circuit is structured decomposable and deterministic.*

This is a rather sobering result, as structured decomposability and determinism are some of the strongest properties that we can impose on a circuit for tractability, and demonstrates the gap between causal and probabilistic inference (which only required decomposability and smoothness). It places causal queries in the small group of queries on a single circuit for which these properties do not suffice; to the best of our knowledge, the only other such type of query studied in literature is the marginal MAP query (MMAP) [9, 3]. In addition, we can view this result as placing a theoretical barrier on interpreting probabilistic circuits as causal models [23, 13]; namely that, even if such an interpretation exists, it is not possible to tractably perform causal inference with the causal PC.

## 4 Symbolic Causal Inference via Operations

Pearl [14] derived a set of rules known as the *do-calculus* that enable one to transform between interventional and observational distributions, by reasoning about the properties of the causal graph. Later, it was shown that the do-calculus is complete; that is, for any identifiable causal effect, it is

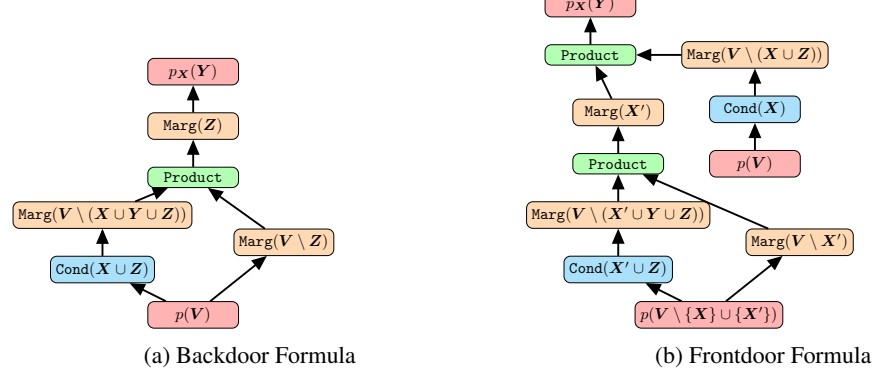

(a) Backdoor Formula          (b) Frontdoor Formula

Figure 2: Compositional pipelines for the backdoor and frontdoor formulae

possible to derive a formula for the effect via do-calculus [19, 10]. In particular, the *ID* algorithm derives a formula by composing the following primitive operations on probability distributions:

**Definition 1** (Marginalization). $\mathtt{MARG}(\cdot; \boldsymbol{W})$ *is a unary operation that takes as input a probability distribution $p$ over variables $\boldsymbol{V}$ and outputs the probability distribution $p(\boldsymbol{V} \setminus \boldsymbol{W}) = \sum_{\boldsymbol{W}} p(\boldsymbol{V})$.*

**Definition 2** (Conditioning). $\mathtt{COND}(\cdot; \boldsymbol{W})$ *is a unary operation that takes as input a probability distribution $p$ over variables $\boldsymbol{V}$ and outputs the probability distribution $p(\boldsymbol{V} \setminus \boldsymbol{W} | \boldsymbol{W}) = \frac{p(\boldsymbol{V})}{p(\boldsymbol{W})}$.*

**Definition 3** (Product). $\mathtt{PROD}(\cdot, \cdot)$ *is a binary operation with parameters that takes as inputs two probability distributions $p, p'$ over variables $\boldsymbol{V}, \boldsymbol{V}'$ and outputs the probability distribution $p \times p'$ over variables $\boldsymbol{V} \cup \boldsymbol{V}'$.*

This can be used to systematically decompose a causal formula into a composition of primitive operations. Such a decomposition can be visualized as a symbolic *compositional pipeline* [21]; for example, in Figure 2, we show pipelines for the backdoor and frontdoor formulae. It can be seen that this provides a sufficient condition for tractability: if we can show that all of the individual operations through the pipeline are tractable for a particular model class, then it follows that the causal formula is tractable. Further, if we have tractable algorithms for each of the operations, then the pipeline provides a tractable algorithm for evaluating the formula. We now show how this framework can be used to analyze the tractability of the backdoor and frontdoor formulae.

**Tractable Conditioning**     Both pipelines require an application of $\mathtt{COND}(\cdot; \boldsymbol{X} \cup \boldsymbol{Z})$ to the observational distribution $p(\boldsymbol{V})$ (and $\mathtt{COND}(\cdot; \boldsymbol{X})$ also for frontdoor), for which no algorithms or tractability results currently exist for PCs. We find that the key property needed for tractability is *strong determinism*[1]. Intuitively, strong determinism requires that, for every sum node $T$, there is a subset of variables $\boldsymbol{W}_L \subset \phi(T)$ such that the value of $\boldsymbol{W}_L$ "determines" which child of $T$ "is active". For example, the root sum node in Figure 3a has scope $\phi(T) = \{X, Y, Z\}$, and $\boldsymbol{W}_L := \{X, Z\}$. Notice that the two children of $T$ partition the possible values of $\boldsymbol{W}_L$. The sum node can thus be interpreted as conditioning on the value of $\boldsymbol{W}_L$.

The canonical example of a strongly deterministic circuit is the probabilistic sentential decision diagram (PSDD) [12], and there exist other circuits which effectively enforce strong determinism, in particular structured decomposable and deterministic XPCs [7]. Crucially, strong determinism is not just a theoretical property; there exist algorithms for learning both PSDDs and such XPCs. We now show that we can tractably apply conditioning to a strongly deterministic PC, but only for some conditioning sets $\boldsymbol{W}$, which depends on its vtree.

**Definition 4** (Tractable Conditioning Sets). *Given a vtree $v$, let $M_v$ be the sequence of vtree nodes obtained by starting at the root and iteratively picking the right child, and define $M_v(i)$ to be the subsequence of $M_v$ consisting of the first $i$ elements. Then we define $\mathcal{Q}(v) = \{\bigcup_{m \in M_v(i)} \phi(m_L) | i = 1, ..., |M_v|\}$ to be the tractable conditioning sets for $v$.*

---

[1]We provide a formal definition of strong determinism in Appendix B.2.

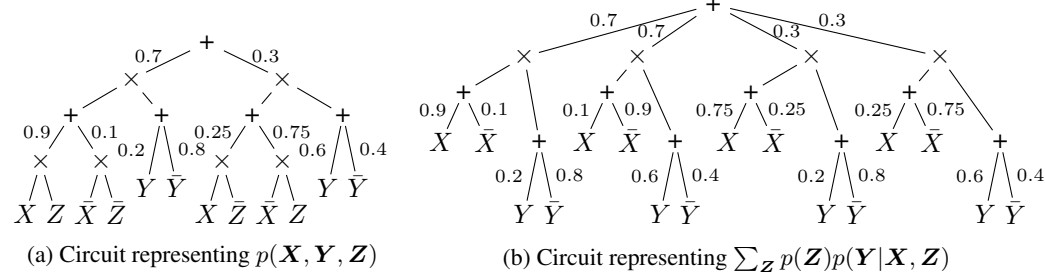

(a) Circuit representing $p(\boldsymbol{X}, \boldsymbol{Y}, \boldsymbol{Z})$          (b) Circuit representing $\sum_{\boldsymbol{Z}} p(\boldsymbol{Z}) p(\boldsymbol{Y}|\boldsymbol{X}, \boldsymbol{Z})$

Figure 3: Example of a structured decomposable and strongly deterministic circuit, and the circuit that results from applying backdoor adjustment as in Theorem 2.

**Proposition 1** (Tractable Conditioning). *Given a structured decomposable (respecting vtree $v$), smooth and strongly deterministic circuit $C$, then if $\boldsymbol{W} \in \mathcal{Q}(v)$, $\texttt{COND}(C; \boldsymbol{W})$ can be computed in $O(|C|)$ time as a structured decomposable (respecting $v$), smooth and deterministic circuit.*

**Tractable Marginals and Products**     Having resolved the conditioning operation, we now turn to the marginalization and product operations.

**Proposition 2** (Tractable Marginals). *Given a structured decomposable (respecting vtree $v$) and smooth circuit $C$, then $\texttt{MARG}(C; \boldsymbol{W})$ can be computed in $O(|C|)$ time as a structured decomposable (respecting vtree $v$) circuit.*

**Proposition 3** (Tractable Products). *[18, 21] Given two structured decomposable (both respecting vtrees $v$) and smooth circuits $C_1, C_2$, then $\texttt{PROD}(C_1, C_2)$ can be computed in $O(|C_1||C_2|)$ time as a structured decomposable (respecting vtree $v$) circuit.*

Looking at the pipelines as a whole, the previous results can be combined to characterize the conditions for tractability and complexity of computation for the backdoor and frontdoor formulae:

**Theorem 2** (Tractable Backdoor and Frontdoor Adjustment). *Suppose that we have a circuit $C$ representing $p(\boldsymbol{V})$ that is structured decomposable (respecting vtree $v$) and strongly deterministic such that $\boldsymbol{X} \cup \boldsymbol{Z} \in \mathcal{Q}(v)$. Then the backdoor formula can be computed exactly in $O(|C|^2)$, and if additionally $\boldsymbol{Z} \in \mathcal{Q}(v)$, the frontdoor formula can be computed exactly in $O(|C|^3)$ time.*

In Figure 3, we can apply backdoor adjustment to the structured decomposable and strongly deterministic circuit in Figure 3a since $\{X, Z\}$ is a tractable conditioning set. The resulting circuit in Figure 3b represents the interventional distribution $p_X(Y)$ in the backdoor case.

We conclude this section with two comments. Firstly, if we are interested in backdoor/frontdoor computation for specific sets $\boldsymbol{X}, \boldsymbol{Z}$, then we will have to choose the vtree such that the tractability conditions hold, though we have some flexibility in this choice. Secondly, we note that the output of the algorithm (e.g. Figure 3b) is actually a structured-decomposable and smooth probabilistic circuit representing $p_{\boldsymbol{X}}(\boldsymbol{Y})$. That is, we obtain a tractable circuit that enables evaluation for *arbitrary* values of $\boldsymbol{X}, \boldsymbol{Y}$, as well as *arbitrary* marginal and conditional inference on the interventional distribution.

## 5   Conclusion

Recent advances in the modelling capabilities of probabilistic circuits have opened up the alluring possibility of tractable high-dimensional causal inference. However, in contrast to probabilistic inference, the theoretical basis for exact causal inference on circuits has remained unclear. We provide the first theoretical results in this direction, showing that, unfortunately, many commonly used classes of probabilistic circuits are insufficient. Nonetheless, for strongly deterministic circuits, we show that the situation is different and derive, remarkably, *polynomial-time* algorithms for backdoor and frontdoor adjustment. At the core of our approach is the extensible representation of causal formulae as a compositional pipeline, allowing us to reduce the analysis to primitive operations. We thus hope that our approach will provide a foundation for future work on tractable causal inference, which could examine tractability for broader classes of causal formulae or probabilistic circuits, or design practical schemes for performing causal inference from data.

## Acknowledgements

This project was funded by the ERC under the European Union's Horizon 2020 research and innovation programme (FUN2MODEL, grant agreement No.834115).

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

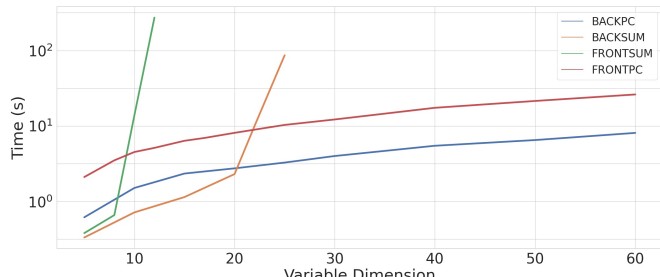

Figure 4: Computation time for backdoor and frontdoor formulae for probabilistic circuits

## A Empirical Validation

We perform preliminary experiments to validate the performance of the following task: given a probabilistic circuit representing some distribution $p(\boldsymbol{V})$, compute $p_{\boldsymbol{x}}^{\text{back}}(\boldsymbol{y}) = \sum_{\boldsymbol{Z}} p(\boldsymbol{Z}) p(\boldsymbol{y}|\boldsymbol{x}, \boldsymbol{Z})$ and $p_{\boldsymbol{x}}^{\text{front}}(\boldsymbol{y}) = \sum_{\boldsymbol{Z}} p(\boldsymbol{Z}|\boldsymbol{x}) \sum_{\boldsymbol{X}'} p(\boldsymbol{X}') p(\boldsymbol{y}|\boldsymbol{X}', \boldsymbol{Z})$ for specific values $\boldsymbol{x}, \boldsymbol{y}$. We compare our algorithms, which we call BACKPC and FRONTPC, to the naïve exact algorithms which compute the backdoor/frontdoor formula by explicitly performing the summations, which we call BACKSUM and FRONTSUM. In Figure 4, we see that, as the dimension of the variables increases, BACKPC and FRONTPC perform the exact computation much more efficiently than the explicit methods, as expected.

## B Proofs

### B.1 Proof of Theorem 1

**Theorem 1.** *The backdoor query for decomposable and smooth circuits is #P-hard, even if the circuit is structured decomposable and deterministic.*

*Proof.* We prove this in the case of binary variables for brevity of presentation, though the proof can easily be extended to non-binary discrete variables. Our proof is based on a reduction from the problem of computing the expectation of a logistic regression model, which was defined and shown to be #P-hard in Van den Broeck et al. [20] and which we refer to as the EXPLR problem. In particular, for any EXPLR problem over variables $\boldsymbol{Z}$, with input size $n_{\boldsymbol{Z}} = |\boldsymbol{Z}|$, we construct a circuit in time and with size linear in $\boldsymbol{Z}$ and where computing the backdoor query is equivalent to solving the EXPLR problem.

The EXPLR problem is defined as computing the following quantity (where $w_i \in \mathbb{R}$):

$$\sum_{\boldsymbol{z}} \frac{1}{1 + e^{-(w_0 + \sum_i w_i z_i)}} \tag{2}$$

We will construct a circuit over variables $\boldsymbol{V} = \{\boldsymbol{X}, \boldsymbol{Y}, \boldsymbol{Z}\}$, where the sets $\boldsymbol{X} = \{X\}$ and $\boldsymbol{Y} = \{Y\}$ each consist of a single variable, and consider the backdoor query for instantiations $x, y$ of $X, Y$. We begin by defining a number of auxiliary circuits/nodes for $\boldsymbol{X}, \boldsymbol{Y}$ and $\boldsymbol{Z}$ individually, all structured decomposable, smooth and deterministic, which will be part of the construction of the main circuit.

First, for $\boldsymbol{Y}$ we define the leaf nodes $\mathbb{1}_y(Y) := \mathbb{1}(Y = y)$ and $\mathbb{1}_{\neg y}(Y) := \mathbb{1}(Y = \neg y)$. We then define $\mathbb{1}_x(X) := \mathbb{1}(X = x)$ and $\mathbb{1}_{\neg x}(X) := \mathbb{1}(X = \neg x)$ for $\boldsymbol{X}$ in a similar manner. Finally, for $\boldsymbol{Z}$, we define two circuits, $\mathbb{1}_{\boldsymbol{Z}}(\boldsymbol{Z})$ and $C_{\boldsymbol{Z}}$, as follows. Let $\boldsymbol{Z} := \{Z_1, ..., Z_{n_{\boldsymbol{Z}}}\}$ be an arbitrary of the variables in $\boldsymbol{Z}$, and let $\boldsymbol{Z}_{\geq i}$ denote $\{X_i, ..., X_{n_{\boldsymbol{Z}}}\}$ for any $1 \leq i \leq n_{\boldsymbol{Z}}$. Then we define the circuit $\mathbb{1}_{\boldsymbol{Z}}(\boldsymbol{Z})$ recursively as follows, where $\mathbb{1}_{\boldsymbol{Z}}(\boldsymbol{Z}) := \mathbb{1}_{\boldsymbol{Z}_{\geq 1}}(\boldsymbol{Z}_{\geq 1})$:

$$\mathbb{1}_{\boldsymbol{Z}_{\geq i}}(\boldsymbol{Z}_{\geq i}) := \begin{cases} \mathbb{1}_{Z_i}(Z_i) \times \mathbb{1}_{\boldsymbol{Z}_{\geq i+1}}(\boldsymbol{Z}_{\geq i+1}) & 1 \leq i < n_{\boldsymbol{Z}} \\ \mathbb{1}_{Z_i}(Z_i) & i = n_{\boldsymbol{Z}} \end{cases} \tag{3}$$

This circuit consists of a series of product units, and leaf units for each $Z_i \in Z$ which are defined to take the value $\mathbb{1}_{Z_i}(z_i) \equiv 1$ for *all* values $z_i$ of $Z_i$. Thus, the circuit as a whole satisfies $\mathbb{1}_{\mathbf{Z}}(\mathbf{Z}) \equiv 1$ for all values $\mathbf{z}$ of $\mathbf{Z}$. In terms of structural properties, the circuit is trivially deterministic and smooth as it does not contain any sum nodes, and is clearly also structured decomposable. Finally, it can also be seen that the size $|\mathbb{1}_{\mathbf{Z}}(\mathbf{Z})|$ (number of edges) of $\mathbb{1}_{\mathbf{Z}}(\mathbf{Z})$ is $O(n_{\mathbf{Z}})$.

We now design a circuit $C_{\mathbf{Z}}(\mathbf{Z})$ to represent the function $e^{-(w_0 + \sum_i w_i z_i)}$.

$$C_{\mathbf{Z} \geq i}(\mathbf{Z} \geq i) := \begin{cases} C_{Z_i}(Z_i) \times C_{\mathbf{Z}_{\geq i+1}}(\mathbf{Z}_{\geq i+1}) & 1 \leq i < n_{\mathbf{Z}} \\ C_{Z_i}(Z_i) & i = n_{\mathbf{Z}} \end{cases} \tag{4}$$

where we define leaf nodes $C_{Z_i}(Z_i) = e^{-w_i z_i}$ for $1 \leq i < n_{\mathbf{Z}}$ and $e^{-(w_0 + w_i z_i)}$ for $i = n_{\mathbf{Z}}$. This circuit is deterministic and smooth, and also decomposes in the same way as $\mathbb{1}_{\mathbf{Z}}(\mathbf{Z})$, i.e. they are structured decomposable with the same vtree. It can also be seen that the size $|C_{\mathbf{Z}}(\mathbf{Z})|$ of $C_{\mathbf{Z}}(\mathbf{Z})$ is $O(n_{\mathbf{Z}})$.

Now, consider the probabilistic circuit over $\mathbf{V} = \mathbf{X} \cup \mathbf{Y} \cup \mathbf{Z}$ given by

$$C(\mathbf{V}) = \mathbb{1}_y(Y) \times (\mathbb{1}_x(X) \times \mathbb{1}_{\mathbf{Z}}(\mathbf{Z}) + \mathbb{1}_{\neg x}(X) \times \mathbb{1}_{\mathbf{Z}}(\mathbf{Z})) \tag{5}$$
$$+ \mathbb{1}_{\neg y}(Y) \times (\mathbb{1}_x(X) \times C_{\mathbf{Z}}(\mathbf{Z}) + \mathbb{1}_{\neg x}(X) \times \mathbb{1}_{\mathbf{Z}}(\mathbf{Z})) \tag{6}$$

$C$ is structured decomposable as all of the product units with the same scope in the equation above decompose in the same way, and we have seen that $\mathbb{1}_{\mathbf{Z}}(\mathbf{Z})$ and $C_{\mathbf{Z}}(\mathbf{Z})$ are structured decomposable with respect to the same vtree. It is also smooth and deterministic as the individual circuits $\mathbb{1}_{\mathbf{Z}}(\mathbf{Z})$ and $C_{\mathbf{Z}}(\mathbf{Z})$ are smooth deterministic, and the sum nodes in the equation satisfy determinism by the fact that $(\mathbb{1}_y(Y), \mathbb{1}_{\neg y}(Y))$ and $(\mathbb{1}_x(X), \mathbb{1}_{\neg x}(X))$ have disjoint support. Finally, as the sizes of $\mathbb{1}_{\mathbf{Z}}(\mathbf{Z})$ and $C_{\mathbf{Z}}(\mathbf{Z})$ are $O(n_{\mathbf{Z}})$, $|C(\mathbf{V}))|$ is also $O(n_{\mathbf{Z}})$.

Now, we show that the backdoor query on $C(\mathbf{V})$ is equivalent to solving the corresponding EXPLR problem. First, we derive expressions for all of the individual components of the backdoor formula.

$$\begin{aligned} C(x, y, \mathbf{z}) &= \mathbb{1}_y(y) \times \mathbb{1}_x(x) \times \mathbb{1}_{\mathbf{Z}}(\mathbf{z}) \\ &= 1 \\ C(x, \mathbf{z}) &= \sum_{y'} C(x, y', \mathbf{z}) \\ &= \mathbb{1}_y(y) \times \mathbb{1}_x(x) \times \mathbb{1}_{\mathbf{Z}}(\mathbf{z}) + \mathbb{1}_{\neg y}(\neg y) \times \mathbb{1}_x(x) \times C_{\mathbf{Z}}(\mathbf{z}) \\ &= 1 + C_{\mathbf{Z}}(\mathbf{z}) \end{aligned}$$

$$\begin{aligned} C(\mathbf{z}) &= \sum_{x'} C(x', \mathbf{z}) \\ &= C(x, \mathbf{z}) + \sum_{y'} C(\neg x, y', \mathbf{z}) \\ &= C(x, \mathbf{z}) + \mathbb{1}_y(y) \times \mathbb{1}_{\neg x}(\neg x) \times \mathbb{1}_{\mathbf{Z}}(\mathbf{z}) + \mathbb{1}_{\neg y}(\neg y) \times \mathbb{1}_{\neg x}(\neg x) \times \mathbb{1}_{\mathbf{Z}}(\mathbf{z}) \\ &= C(x, \mathbf{z}) + 2 \end{aligned}$$

The backdoor query for $C$ can then be expressed as

$$\begin{aligned} \sum_{\mathbf{z}} C(\mathbf{z}) C(y | x, \mathbf{z}) &= \sum_{\mathbf{z}} (C(x, \mathbf{z}) + 2) \frac{C(x, y, \mathbf{z})}{C(x, \mathbf{z})} \\ &= \sum_{\mathbf{z}} \left[ 1 + \frac{2}{1 + C_{\mathbf{Z}}(\mathbf{z})} \right] \\ &= 2^{n_{\mathbf{Z}}} + 2 \sum_{\mathbf{z}} \frac{1}{1 + e^{-(w_0 + \sum_i w_i z_i)}} \end{aligned}$$

Thus, if we can compute the backdoor query for $C$, then we can compute the given EXPLR problem, completing the reduction. $\qquad \square$

**Corollary 1.** *For any causal effect $p_{\boldsymbol{x}}(\boldsymbol{y})$ and causal diagram $G$ such that the query is identifiable through a backdoor adjustment, and the observational distribution $p(\boldsymbol{V})$ given as a decomposable and smooth circuit $C(\boldsymbol{V}) \equiv p(\boldsymbol{V})$, computing $p_{\boldsymbol{x}}(\boldsymbol{y})$ is #P-hard, even if the circuit is structured decomposable and deterministic.*

*Proof.* By the identifiability condition, we have that $p_{\boldsymbol{x}}(\boldsymbol{y}) = \sum_{\boldsymbol{z}} p(\boldsymbol{y}|\boldsymbol{x}, \boldsymbol{z}) p(\boldsymbol{z}) = \sum_{\boldsymbol{z}} C(\boldsymbol{z}) C(\boldsymbol{y}|\boldsymbol{x}, \boldsymbol{z})$, which is the backdoor query. Hardness of computing the causal effect then follows from hardness of backdoor queries for the probabilistic circuit. $\qquad\square$

## B.2 Proof of Proposition 1

**Proposition 1** (Tractable Conditioning). *Given a structured decomposable (respecting vtree $v$), smooth and strongly deterministic circuit $C$, then if $\boldsymbol{W} \in \mathcal{Q}(v)$, $\texttt{COND}(C; \boldsymbol{W})$ can be computed in $O(|C|)$ time as a structured decomposable (respecting $v$), smooth and deterministic circuit.*

We begin with a more precise definition of strong determinism, which generalizes the definition of Kisa et al. [12] to general probabilistic circuits (rather than just PSDDs). In order to define this, we first note that every sum/leaf node in a structured decomposable and smooth circuit will have a "corresponding" vtree node which has the same scope.

**Lemma 1.** *Let $C$ be a structured decomposable (respecting vtree $v$) and smooth circuit which contains at least one product node. Then for every sum or leaf node $N$ in the circuit, there exists a vtree node $m$ in $v$ such that $\phi(N) = \phi(m)$.*

*Proof.* For every sum/leaf node which has a product node parent, this follows by the definition of structured decomposability. If the root is a sum node, then by smoothness we can note that all descendants of the root "before reaching a product node" must have the same scope, and so the root itself must have the same scope as any reached product nodes (by assumption, there exists at least one product node in the circuit). Finally, for every sum/leaf node which has a sum node parent, then the result follows by iteratively considering its parent, until we reach a sum node with a product node parent, or the root. $\qquad\square$

**Definition 5** (Strong Determinism). *A structured decomposable and smooth circuit (w.r.t. vtree $v$) is strongly deterministic if for every sum node $T$, and for every instantiation $\boldsymbol{w}_L$ of the scope of the left child of the corresponding vtree node, at most one of the children $N_i$ of $T$ evaluates to a non-zero value $N_i(\boldsymbol{w}_L) > 0$.*

Intuitively, strong determinism implies that each child of a sum node $T$ has disjoint support over $\boldsymbol{W}_L \subset \phi(T)$. In the case of PSDDs, this is achieved by assigning to each child mutually exclusive logical formulae over $\boldsymbol{W}_L$ (which are themselves represented by PSDDs). Meanwhile, in the case of structured decomposable and deterministic XPCs [7], this is achieved by assigning to each child distinct logical conjunctions over the variables in $\boldsymbol{W}_L$.

Now, recall the definition of the tractable conditioning sets $\mathcal{Q}(v)$:

**Definition 4** (Tractable Conditioning Sets). *Given a vtree $v$, let $M_v$ be the sequence of vtree nodes obtained by starting at the root and iteratively picking the right child, and define $M_v(i)$ to be the subsequence of $M_v$ consisting of the first $i$ elements. Then we define $\mathcal{Q}(v) = \{\bigcup_{m \in M_v(i)} \phi(m_L) | i = 1, ..., |M_v|\}$ to be the tractable conditioning sets for $v$.*

Intuitively, the reason for this definition is that if $\boldsymbol{W} = \bigcup_{m \in M_v(i)} \phi(m_L) \in \mathcal{Q}(v)$, then for any instantiation $\boldsymbol{w}$ of $\boldsymbol{W}$, strong determinism ensures that for any sum node corresponding to a vtree node in $M_v(i)$ only one child is "active". This allows us to interpret such sum nodes as conditioning on $\boldsymbol{W}$. In fact, we can extend this to all sum/leaf nodes, which either also have only one active child under $\boldsymbol{w}$, or else have scope entirely disjoint from $\boldsymbol{W}$ and thus are essentially not affected by conditioning on $\boldsymbol{W}$.

**Lemma 2.** *Let $C$ be a structured decomposable PC respecting $v$, and suppose $\boldsymbol{W}$ is an element of $\mathcal{Q}(v)$. Let $N$ be any sum/leaf node in $C$, with corresponding vtree node $m$. Then it holds that either $\phi(m_L) \subseteq \boldsymbol{W}$ (or $\phi(m) \subseteq \boldsymbol{W}$ if $m$ is a leaf in the vtree), or else $\phi(N) \cap \boldsymbol{W} = \emptyset$.*

*We define $\boldsymbol{N}_{\boldsymbol{W}}$ to be the set of all sum/leaf units satisfying the former condition, and $\boldsymbol{N}_{\neg\boldsymbol{W}}$ the latter.*

*Proof.* Since $\boldsymbol{W} \in \mathcal{Q}(v)$, it is a union of the scopes of the left children of $M_v(i)$ for some $i$. Define $M_{v,L}(i) := \{m_L : m \in M_v(i)\}$ to be the set of all such left children. Also, define $m_{v,R}(i)$ to be the right child of the last ($i^{\text{th}}$) node in the sequence $M_v(i)$, which has scope $\phi(m_{v,R}(i)) = \boldsymbol{V} \setminus \bigcup_{m \in M_v(i)} \phi(m_L) = \boldsymbol{V} \setminus \boldsymbol{W}$.

Now, every node $m$ in the vtree is either in $M_v(i)$, a descendant of some node in $M_{v,L}(i)$, or a descendant of $m_{v,R}(i)$. In the first case, $\phi(m_L) \subseteq \bigcup_{m \in M_v(i)} \phi(m_L) = \boldsymbol{W}$ follows by definition. In the second case, note that $\phi(m) \subseteq \boldsymbol{W}$ as it is a descendant of some left child of $M_v(i)$, so $\phi(m_L) \subset \phi(m) \subseteq \boldsymbol{W}$ follows (or, if $m$ is a leaf in $v$, $\phi(m) \subseteq \boldsymbol{W}$ holds). Finally, in the third case, $m$ is a descendant of $m_{v,R}(i)$, so $\phi(m) \subseteq \boldsymbol{V} \setminus \boldsymbol{W}$ which implies $\phi(m) \cap \boldsymbol{W} = \emptyset$. The lemma then follows by the fact that every sum/leaf node has a corresponding vtree node with the same scope. $\square$

We use this characterization to define an operation on the input circuit $C$, which we will show faithfully represents the conditioning operation.

**Definition 6** (Conditional Circuit). *Let $C$ be a structured decomposable (respecting vtree $v$) and smooth PC, and let $\boldsymbol{W} \in \mathcal{Q}(v)$. We define the conditional circuit $C_{cond(\boldsymbol{W})}$ of $C$ to a copy of $C$, with the following changes:*

- *For every sum node $T \in \boldsymbol{N_W}$, set $\theta_i' := 1$ for all weights $\theta_i'$ of the new sum node $T'$.*

- *For every leaf node $L \in \boldsymbol{N_W}$, set $L'(\phi(L)) \equiv \mathbb{1}_{\phi(L) \in supp(L)}$, where $supp(L) = \{\boldsymbol{l} : L(\boldsymbol{l}) > 0\}$*

- *For every sum node $T \in \boldsymbol{N_{\neg W}}$, set $\theta_i' := \frac{\theta_i N_i(\emptyset)}{\sum_j \theta_j N_j(\emptyset)}$ for all weights $\theta_i'$ of the new sum node $T'$ (i.e. normalize the weights).*

- *For every leaf node $L \in \boldsymbol{N_{\neg W}}$, set $L'(\phi(L)) \equiv \frac{L(\phi(L))}{\sum_{\phi(L)} L(\phi(L))}$ (i.e. normalize the leaf nodes).*

We now prove the proposition by showing that $C_{cond(\boldsymbol{W})}$ is indeed a circuit satisfying the required conditions.

*Proof. (of Proposition)* Firstly, it is clear that we can compute the conditional circuit for $C$ in $O(|C|)$ time, using a single forward pass through the circuit starting at the leaves (and keeping track of the value of $N(\emptyset)$ at each node $N$). Further, $C_{cond(\boldsymbol{W})}$ is structured decomposable (respecting the same vtree) and smooth as the operations do not change the scope of any of the nodes, and is further strongly deterministic as the support of the leaf nodes do not change in the operation.

It remains to show that $C_{cond(\boldsymbol{W})}$ faithfully represents the conditional, i.e. $C_{cond(\boldsymbol{W})}(\boldsymbol{V}) \equiv \frac{C(\boldsymbol{V})}{C(\boldsymbol{W})}$. Let $\boldsymbol{v}$ be an instantiation of $\boldsymbol{V}$, and $\boldsymbol{w}$ the part of $\boldsymbol{v}$ corresponding to $\boldsymbol{W}$. We prove this by induction; namely, by showing that $N'(\phi(N)) = \frac{N(\phi(N))}{N(\boldsymbol{w} \cap \phi(N))}$ holds for all nodes $N$ in $C$ (and their corresponding node $N'$ in $C_{cond(\boldsymbol{W})}$).

For the base case, we consider leaf nodes. For any leaf node $L \in \boldsymbol{N_W}$ (such that $\phi(L) \subseteq \boldsymbol{W}$), we have that $\frac{L(\phi(L))}{L(\boldsymbol{w} \cap \phi(L))} = \frac{L(\boldsymbol{w} \cap \phi(L))}{L(\boldsymbol{w} \cap \phi(L))} = 1 = L'(\phi(L))$, as required. On the other hand, if $L \in \boldsymbol{N_{\neg W}}$ (such that $\phi(L) \cap \boldsymbol{W} = \emptyset$), then $\frac{L(\phi(L))}{L(\boldsymbol{w} \cap \phi(L))} = \frac{L(\phi(L))}{L(\emptyset)} = \frac{L(\phi(L))}{\sum_{\phi(L)} L(\phi(L))} = L'(\phi(L))$.

Now, consider sum nodes $T$. For any sum node $T \in \boldsymbol{N_W}$, we know that $\phi(m_L) \subseteq \boldsymbol{W}$ and thus, by strong determinism, only one child $N_i$ has a non-zero value under the instantiation $\boldsymbol{w}$. Let $\theta_i$ be the corresponding weight for this child in $C$. Then we have that $\frac{T(\phi(T))}{T(\boldsymbol{w} \cap \phi(T))} = \frac{\theta_i N_i(\phi(N_i))}{\theta_i N_i(\boldsymbol{w} \cap \phi(N_i))} = N_i'(\phi(N_i')) = \theta_i' T'(\phi(T')) = T'(\phi(T'))$. Here, the second equality follows by the inductive hypothesis on $N_i'$, while the third inequality follows as we have shown that the conditional circuit retains strong determinism. Now, for sum nodes $T \in \boldsymbol{N_{\neg W}}$, such that $\phi(T) \cap \boldsymbol{W} = \emptyset$, we have $\frac{T(\phi(T))}{T(\boldsymbol{w} \cap \phi(T))} = \frac{\sum_i \theta_i N_i(\phi(N_i))}{\sum_j \theta_j N_j(\boldsymbol{w} \cap \phi(N_j))} = \frac{\sum_i \theta_i N_i(\phi(N_i))}{\sum_j \theta_j N_j(\emptyset)} = \sum_i \theta_i' \frac{N_i(\phi(N_i))}{N_i(\emptyset)} = \sum_i \theta_i' \frac{N_i(\phi(N_i))}{N_i(\boldsymbol{w} \cap \phi(N_i))} = \sum_i \theta_i' N_i'(\boldsymbol{w} \cap \phi(N_i')) = T'(\phi(T'))$.

Finally, consider product nodes $P$, which are not directly changed in the conditional circuit. If $N_i'$ are the children of the new product node $P'$, we have that $P'(\phi(P)) = \prod_i N_i'(\phi(N_i)) =$

$\prod_i \frac{N_i(\phi(N_i))}{N_i(\boldsymbol{w} \cap \phi(N_i))} = \frac{\prod_i N_i(\phi(N_i))}{\prod_i N_i(\boldsymbol{w} \cap \phi(N_i))}$ by the inductive hypothesis. The numerator $\prod_i N_i(\phi(N_i)) = P(\phi(P))$ by definition, while the denominator $\prod_i N_i(\boldsymbol{w} \cap \phi(N_i)) = \prod_i \sum_{\phi(N_i) \setminus \boldsymbol{w}} N_i(\phi(N_i)) = \sum_{\bigcup_i \phi(N_i) \setminus \boldsymbol{w}} \prod_i N_i(\phi(N_i)) = \sum_{\phi(P) \setminus \boldsymbol{w}} P(\phi(P)) = P(\boldsymbol{w} \cap \phi(P))$, where, crucially, we can interchange the sum and product because $\phi(N_i) \cap \phi(N_j) = \emptyset$ for any $i \neq j$ by decomposability. This gives $P'(\phi(P)) = \frac{P(\phi(P))}{P(\boldsymbol{w} \cap \phi(P))}$, as required.

Thus, we have shown that $N'(\phi(N)) = \frac{N(\phi(N))}{N(\boldsymbol{w} \cap \phi(N))}$ for all nodes, and in particular, for the root, we have that $C_{\mathrm{cond}(\boldsymbol{W})}(\boldsymbol{V}) \equiv \frac{C(\boldsymbol{v})}{C(\boldsymbol{w})}$. This concludes the proof.

$\square$

## B.3   Proof of Proposition 2

**Proposition 2** (Tractable Marginals). *Given a structured decomposable (respecting vtree $v$) and smooth circuit $C$, then* MARG$(C; \boldsymbol{W})$ *can be computed in* $O(|C|)$ *time as a structured decomposable (respecting vtree $v$) circuit.*

*Proof.* The tractability of the marginalization operation essentially follows from the procedure for marginal inference in decomposable and smooth circuits; namely, to sum out $\boldsymbol{W}$ by replacing each leaf $L(\phi(L))$ with $L'(\phi(L) \setminus \boldsymbol{W}) := \sum_{\phi(L) \cap \boldsymbol{W}} L(\phi(L))$. However, this does not preserve structured decomposability with the same vtree (as the scopes of the leaf nodes and thus product nodes will change). To resolve this, we use a trick: we define $L'$ to instead be a function over $\phi(L)$, whose value dues not depend on the variables $\phi(L) \cap \boldsymbol{W}$. This ensures that the scopes of all nodes remains the same, such that the output circuit is structured decomposable with respect to the same vtree.  $\square$

## B.4   Proof of Theorem 2

**Theorem 2** (Tractable Backdoor and Frontdoor Adjustment). *Suppose that we have a circuit $C$ representing $p(\boldsymbol{V})$ that is structured decomposable (respecting vtree $v$) and strongly deterministic such that $\boldsymbol{X} \cup \boldsymbol{Z} \in \mathcal{Q}(v)$. Then the backdoor formula can be computed exactly in $O(|C|^2)$, and if additionally $\boldsymbol{Z} \in \mathcal{Q}(v)$, the frontdoor formula can be computed exactly in $O(|C|^3)$ time.*

*Proof.* In Figure 2, we see that both the backdoor and frontdoor formulae consist of conditioning operations, followed by compositions of marginalization and product operations. Given that $\boldsymbol{X} \cup \boldsymbol{Z} \in \mathcal{Q}(\boldsymbol{Z})$, then by Proposition 1 we can compute COND$(C; \boldsymbol{X} \cup \boldsymbol{Z})$ in $O(|C|)$ time, outputting a structured decomposable circuit that is of $O(|C|)$ size. A similar argument applies for the COND$(C; \boldsymbol{X} \cup \boldsymbol{Z})$ and COND$(C; \boldsymbol{X})$ operations, with the output of both operations respecting the same vtree.

The marginalization operations output a circuit of $O(|C|)$ size respecting the same vtree as the input, so we can discount them in our analysis. Finally, we note that the inputs to every product operation are structured decomposable circuits with the same vtree, so the product operation is tractable in $O(|C_1||C_2|)$. The backdoor pipeline contains just a single product, while the frontdoor pipeline consists of two products, leading to $O(|C|^2)$ and $O(|C|^3)$ complexity respectively.

The algorithm for computing the backdoor/frontdoor formulae follows simply by implementing the COND, MARG and PROD operations (c.f. [21] for PROD), and composing the operations as shown in the pipeline.  $\square$

