# OpenReview forum: "Symbolic Causal Inference via Operations on Probabilistic Circuits"
_NeurIPS.cc/2022/Workshop/nCSI — nCSI WS @ NeurIPS 2022 Poster_

### Official Review · Reviewer_93pp · 2022-10-13
**Overall interesting proposal but with crucial typos making it very difficult to assess corectness of claims.**

**Rating:** 1
**Confidence:** 1

**Review:**

The authors consider the problem of deriving a condition on probabilistic circuits under which causal queries remain tractable. Specifically, they first show that even under the assumptions of structured decomposability and determinism the backdoor query remains non-tractable. They then propose a new condition called strong determinism under and show that under this assumption the backdoor query is tractable for certain backdoor sets.

The problem of causal queries in probabilistic queries is very interesting and of some practical relevance. Therefore, the presented results would be relevant and interesting if correct. It is however, difficult to evaluate the results since the crucial property of strong determinism is introduced only in the Appendix where the definition contains a typo in a crucial sentence. As a result, it is difficult to understand the definition and therefore evaluate the correctness of the results. In addition I also have two more minor comments:
- The L in the subset notation is nowhere defined I think (although I suppose it means the scope of the left child in most places).
- Is there some simple intuition on why the backdoor query is in general non-tractable even if computing conditional probabilities is given that it seems to mostly require computing conditional probabilities?

---

### Official Review · Reviewer_SU7Q · 2022-10-16
**Solid workshop contribution on the computational tractability of backdoor and frontdoor adjustment for probabilistic circuits.**

**Rating:** 2
**Confidence:** 2

**Review:**

[Disclaimer: The reviewer has expertise in causality but little to no familiarity with probabilistic circuits.]

**Summary.** The paper investigates the computational tractability of backdoor (BD) and frontdoor (FD) adjustment, that is, given a causal estimand such as the BD or FD formulae involving sums and products of marginal and conditional distributions, under which conditions can they be efficiently computed? To model the (observational) joint distribution of all variables, the paper focuses on the model class of probabilistic circuits (PC), for which tractability of *probabilistic* inference is well-studied. The paper then proceeds to provide some theoretical insights on tractability of PCs for *causal* inference. In particular, as an initial negative result it is shown that BD adjustment is #P hard, even for a fairly restricted class of PCs. The paper then defines additional conditions (for tractable conditioning) and proves that these are sufficient for tractably performing BD and FD adjustment. Very preliminary simulations are consistent with this theoretical insight.

**Evaluation.** This is an interesting and very well-written paper. It studies a question (tractability of computing well-known identifying formulae) which is certainly important, but to which AFAIK not much attention was previously dedicated. The paper provides some theoretical insights into the fundamental limits of this task, and specifies clear conditions under which it becomes possible. Overall, I think the paper is a good fit for the workshop and will be a good addition. I therefore recommend acceptance.

**Questions:**
- It seems that Thm. 1 only applies to BD adjustment. Do you also have a similar negative result to Thm 1. for FD adjustment? If not, why?

**Comments/suggestions:**
- The presentation of the scope of causal inference in the Abstract and Introduction (inferring causal effects from observational data) is slightly inaccurate as this is only a special case. It would be more general and accurate to speak of identifying and estimating causal queries of interest based on assumptions (e.g., causal diagram, but this could also involve, e.g., parametric assumptions) and the *available* data (this does not have to be purely observational, but can also contain mixtures of data from different interventional distributions).
- The quantity $p_X(Y)$ is *not* a causal effect (l.24), it is an interventional distribution. Average causal effects typically take the form of an expectation w.r.t. an interventional distribution, e.g., $E[Y | do(X) ]$.
- Depending on the target audience, especially if also geared towards causality researchers with interest in tractable inference, it would be helpful to provide a more detailed, less technical and more intuitive account of PCs, e.g., using additional figures. This would make the paper more accessible.
- Note: when considering adjustment for sets of variables, in particular for multi-variate treatments $\mathbf{x}$ as done in section 3, care needs to be taken, as some subleties arise w.r.t. proper causal paths. I believe the original backdoor adjustment criterion only applies to singleton treatments, though I'm not 100% sure about this. Worth double checking though.

---

### Meta-Review · Area_Chair_K6Xb · 2022-10-19

**Recommendation:** 3
**Confidence:** 3

**Metareview:**

This paper looks at causal operations and their interoperability with tractable models, making it a good fit for the workshop. they consider how the backdoor and front door adjustment can be phrased in tractable models, and provide some proofs about when this is tractable.

all around the paper is very relevant and I recommend accept. R2 has recommended it get rejected owing to an incorrect definition of strong determinism - I agree that they don't make it precise in some places, so I would recommend clear definitions for all terms - but I don't think they made a mistake here (or at least not that I could find). I didn't check the mathematical aspects in every detail, but the approach seems sound and technically correct.

---

### Decision · Program_Chairs · 2022-10-20

Accept (Poster)